# Iris Recognition Method Based on Parallel Iris Localization Algorithm and Deep Learning Iris Verification

**DOI:** 10.3390/s22207723

**Published:** 2022-10-12

**Authors:** Yinyin Wei, Xiangyang Zhang, Aijun Zeng, Huijie Huang

**Affiliations:** 1Shanghai Institute of Optics and Fine Mechanics, Chinese Academy of Sciences, Shanghai 201800, China; 2Center of Materials Science and Optoelectronics Engineering, University of Chinese Academy of Sciences, Beijing 100049, China; 3School of Science, Jiangnan University, Wuxi 214122, China

**Keywords:** iris recognition, iris localization, iris verification, deep residual network, residual pooling layer

## Abstract

Biometric recognition technology has been widely used in various fields of society. Iris recognition technology, as a stable and convenient biometric recognition technology, has been widely used in security applications. However, the iris images collected in the actual non-cooperative environment have various noises. Although mainstream iris recognition methods based on deep learning have achieved good recognition accuracy, the intention is to increase the complexity of the model. On the other hand, what the actual optical system collects is the original iris image that is not normalized. The mainstream iris recognition scheme based on deep learning does not consider the iris localization stage. In order to solve the above problems, this paper proposes an effective iris recognition scheme consisting of the iris localization and iris verification stages. For the iris localization stage, we used the parallel Hough circle to extract the inner circle of the iris and the Daugman algorithm to extract the outer circle of the iris, and for the iris verification stage, we developed a new lightweight convolutional neural network. The architecture consists of a deep residual network module and a residual pooling layer which is introduced to effectively improve the accuracy of iris verification. Iris localization experiments were conducted on 400 iris images collected under a non-cooperative environment. Compared with its processing time on a graphics processing unit with a central processing unit architecture, the experimental results revealed that the speed was increased by 26, 32, 36, and 21 times at 4 different iris datasets, respectively, and the effective iris localization accuracy is achieved. Furthermore, we chose four representative iris datasets collected under a non-cooperative environment for the iris verification experiments. The experimental results demonstrated that the network structure could achieve high-precision iris verification with fewer parameters, and the equal error rates are 1.08%, 1.01%, 1.71%, and 1.11% on 4 test databases, respectively.

## 1. Introduction

Biometrics methods, which identify people based on physiological or behavioral characteristics [1], have been widely used in various commercial [2], civil [3], and forensic applications [4]. As a biometric technology for identification, iris recognition can be used in several crucial places such as border crossings [5], banks [6], private companies [7], institutes, and law enforcement agencies. The iris is the annular region of the human eye between the sclera and pupil, and the complex iris texture with unique information [8] is suitable for personal identification. However, iris images collected under non-cooperative environments often contain various forms of noise, including reflection points, motion blur, and occlusions of glasses, hair, and eyelashes. This presents a challenge for iris recognition.

Recently, deep learning techniques have been widely used in several research fields, such as photonics research [9,10,11], biological imaging [12,13], material science [14], and image super-resolution [15,16], all of which demonstrate the advantages of these techniques. Similarly, the strategy based on deep learning has achieved excellent performance in a series of biometric technology problems, such as emotion recognition [17,18], gait recognition [19], fingerprint recognition [20], and voice signal recognition [21]. This also urges researchers to use deep learning methods to overcome iris recognition problems in non-cooperative environments. Gangwar A [22] proposed DeeplrisNet architecture for iris recognition in the early stage. This scheme is based on the deep architecture and various tricks for successful applications of CNNs, which could achieve outstanding iris recognition performance based on the deep architecture. Wang [23] proposed a capsule network architecture for iris recognition. By docking different outputs of several classical neural network structures with the capsule architecture, the experimental results indicated the feasibility and stability of the approach. Additionally, Minaee S [24] used VGGNet to extract iris deep features to achieve high-precision recognition results. Nguyen [25] studied several pre-trained CNN models, including AlexNet, VGGNet, InceptionNet, ResNet, and DenseNet, to extract off-the-shelf CNN features for accurate iris recognition. Although the above methods could achieve high-precision iris recognition results, they are all at the cost of increasing the complexity of the model, and there is little research on the iris location stage.

He [26] proposed a deep residual network (ResNet) for image recognition. This architecture can effectively increase accuracy and solve the problem of over fitting, which has great advantages for training the smaller iris datasets. Mao [27] introduced a residual pooling layer for texture recognition, which can preserve the spatial information of texture to achieve advanced recognition performance. The iris has obvious texture features, which motivates us to use this architecture for iris recognition. Based on the above, this paper proposes a deep learning architecture combining ResNet and the residual pooling layer for iris verification. The new lightweight deep learning architecture for iris verification is composed of several residual modules. A simple residual pooling layer was added to extract more texture features of the iris to improve the accuracy of iris recognition. With this network architecture, high-precision iris verification could be realized with low model complexity. More importantly, considering that the actual iris recognition optical system captures the original iris image that is not normalized, we used the parallel circular Hough transform to locate the inner boundary of the iris, after which the fast Daugman integro-differential localization algorithm was used to locate the outer boundary based on the inner boundary information. Thus, a fast and effective iris localization scheme in a non-cooperative environment was realized. Then, the normalization step can be realized by using the inner and outer boundary information of the iris, and the input of the iris verification deep learning architecture can be obtained. The primary findings of this study can be summarized as follows:We used a parallel circular transform to locate the inner boundary of the iris, and the fast Daugman integro-differential localization algorithm for locating the outer boundary, which results in effective and fast iris localization.A new lightweight deep learning architecture was developed for iris verification. The network architecture is based on ResNet, which is used to improve the robustness of the iris verification network architecture and the accuracy of the verification results.A residual pooling layer was introduced into the deep learning network architecture for iris recognition to extract more iris texture features.

The remainder of this paper is organized as follows: Section 2 introduces the related research on iris recognition. Section 3 presents details pertaining to the proposed iris localization and verification methods. Section 4 presents, compares, and analyzes the experimental results. Section 5 summarizes the research and concludes the study.

## 2. Related Work

Iris recognition methods can be divided into two categories: classical manual feature engineering and deep learning-based methods. Compared with the classical manual engineering iris recognition scheme, the advantage of the iris recognition scheme based on deep learning is that it can achieve higher recognition accuracy; however, it requires numerous datasets for training to obtain the deep neural network architecture with sufficient complexity.

Daugman [8] proposed an integro-differential operator to precisely locate the inner and outer boundaries of the iris; following this, a rubber sheet model was used for normalizing the region of the iris, a two-dimensional Gabor filter was used to extract the iris features, and the Hamming distance was computed to compare each pair of iris codes. This represents a classical manual feature engineering method for iris recognition, and several subsequent iris recognition-based studies have been conducted on this basis.

The Deep CNN extracts features to encode iris images into feature vectors and then measures the Euclidean distance between them. The smaller the value of the Euclidean distance, the greater the possibility of the images belonging to the same type of iris. Zhang [28] first incorporated maxout units into CNNs for iris representation, and the experimental results revealed that this scheme could learn iris features rapidly, compactly, and discriminatively. Liu proposed a deep learning-based architecture for heterogeneous iris recognition [29], which learned relevant features to obtain the similarity between iris image pairs based on a CNN. Zhao [30] proposed a unified network (UniNet) which is composed of two sub-networks (FeatNet and MaskNet), and an Extended Triplet Loss (ETL) was introduced for effective supervision for learning comprehensive and spatially-corresponding iris features. Wang [31] proposed DRFNet on the basis of UniNet by introducing the residual network learning with dilated convolutional kernels. These methods show state-of-the-art generalization capability and recognition accuracy.

## 3. Materials and Methods

### 3.1. Iris Localization Stage

Traditional iris localization methods can be divided into Daugman’s integro-differential operator-based methods and Wilde’s circular Hough transforms-based methods. However, the integro-differential operator method for iris localization must estimate the center coordinates of the iris region, which is uncertain in a non-cooperative environment. The accuracy of circular Hough transforms is low for outer boundaries owing to the noise in non-cooperative environments, such as reflection points, motion blur, and the occlusion of eyelashes and glasses.

To address these issues, we developed an efficient iris localization method. First, parallel circular Hough transforms were used to identify the inner boundaries of the iris region. Thereafter, as the center of the pupil always lies close to the center of the iris, we used the center of the inner boundary as the center coordinate to identify the outer boundary using the integro-differential operator.

#### 3.1.1. Inner Boundary Localization

Before using the circular Hough transform to locate the inner boundary, a Canny filter was used to reinforce the inner border of the iris. By applying a Canny filter to the iris image, we could obtain a binary image with bright edge points; the Canny filtering process was executed using a central processing unit (CPU). The circular Hough transform was then used to identify the inner boundaries of the iris. To quickly determine the inner boundary of the iris, circular Hough transforms were executed in parallel on graphics processing units (GPUs) with the compute unified device architecture (CUDA). In the application of the circular Hough transform, we considered edge points in the image space as the input and transformed them into the Hough space. The specific coordinate transformation formula used was as follows:(1)a=x+rcosθ
(2)b=y+rsinθ
where x and y denote the horizontal and vertical coordinates of the edge point in the image space, respectively, r denotes the radius of the search circles, and $\theta \in (0^{{}^{\circ}};360^{{}^{\circ}})$. Using the above coordinate transformation formula, a three-dimensional Hough space with the coordinate system (a,b,r) was formed.

For each edge point coordinate (x,y), with a defined search radius r, a circle was constructed. All of the edge points, which represent the circles in the Hough space, were stored, and the inner boundary of the iris could be recognized according to the following formula:(3)$$H(x_{l},y_{l},r)=\sum_{i=1}^{N}h(x_{i},y_{i},x_{l},y_{l},r)$$
where
(4)$$h(x_{i},y_{i},x_{l},y_{l},r)=\left\{\begin{array}{l} 1,iff(x_{i},y_{i},x_{l},y_{l},r)==0\\ 0,otherwise \end{array}\right.$$
(5)$$f(x_{i},y_{i},x_{l},y_{l},r)={\left(x_{i}-x_{l}\right)}^{2}+{\left(y_{i}-y_{l}\right)}^{2}-r^{2}$$

In Equations (3)–(5), $\left(x_{i},y_{i},r\right)$ denotes the coordinate of the Hough space, and $\left(x_{l},y_{l},r\right)$ denotes information pertaining to the target circle of the search.

Figure 1 illustrates the process of transformation from the image space to the Hough space; for each edge point with a defined search radius, r, we drew a circle in the Hough space. The maximum value indicates that the coordinate is located at the intersection of different loops in the Hough space, which corresponds to the center point in the image space.

Noruzi [32] proposed an effective scheme to perform a circular Hough transform on GPUs for iris localization; the specific working process can be described as follows: the objective is to search for possible circles with different radii to obtain the radius of the target circle. Since a circle is composed of several pixels, the operation of pixels by threads exists on each processor. A processor operates on each search radius independently. These operations are performed in parallel, and all search results are gathered. The highest value for each point in the Hough space represents the target circle. For each search radius, r, each CUDA block has numerous threads dedicated to parallel processing, which is a width x height/block size. Each thread calculates the pixel values of each circle in the Hough space, and then, the Hough space is updated with the search radius and pixel information of the edge points in the image space.

#### 3.1.2. Outer Boundary Localization

The integro-differential operator [8] identifies the outer boundary of the iris with maximum intensity variation. To obtain the outer boundary center point, (x,y), and radius, r, the integro-differential operator can be expressed as:(6)$$\max_{\left(r,x_{0},y_{0}\right)}\left|G_{\delta }(r)*\frac{\partial }{\partial r}\oint \frac{I_{o}(x,y)}{2\pi r}ds\right|$$
where $I_{o}(x,y)$ represents the original iris image, ds denotes a differential circular arc with the center coordinate $\left(x_{0},y_{0}\right)$ and radius r, and $G_{\delta }(r)$ denotes a Gaussian smoothing filter with scale δ. This operator identifies an iris circle by searching the maximum value of the contour integral derivative of the normalized iris circular boundary; the path of the contour integral is determined by the center coordinates and the radius, $\left(x_{0},y_{0},r\right)$. The center coordinates and radius of the inner boundary can be obtained based on the circular Hough transform. Therefore, to detect the outer boundary, we consider the center coordinates of the inner boundary and its 20×20 neighborhood as the input center coordinates, $\left(x_{0},y_{0}\right)$, and the range of the radius, r, is set between the radius of the inner boundary and half the size of the iris image. Using this method, the amount of computation is significantly reduced.

### 3.2. Iris Verification Stage

We propose a new lightweight deep learning network architecture for iris verification. The overall architecture of the proposed network is presented in Figure 2. As depicted in the figure, the proposed iris recognition network consists of feature extraction CNNs and a residual pooling layer, where the green rectangle represents the feature extraction CNNs. Specifically, the input is a normalized iris image, which is forwarded by a convolutional layer, a batch normalization layer, a max-pooling layer, four residual modules, and a residual pooling layer.

#### 3.2.1. Residual Modules

Deep CNNs can extract more characteristics from targets compared with CNNs; however, the underlying training process is difficult. A residual network (residual modules) [26] has been proposed to address this issue.

The residual learning unit of the architecture can be described as:(7)$$y=F(x,W_{i})+W_{s}x$$
Here, x and y represent the input and output vectors of the residual learning unit, respectively; $W_{i}$ and $W_{s}$ denote the weights of the corresponding layers, and the function $F(x,W_{i})$ represents residual mapping. The expression F+x represents a shortcut connection and element-wise addition, which renders the optimization of the deep residual learning network easier.

Each residual module in the proposed network satisfies Equation (7), and its residual mapping comprises two 3×3 convolution and batch normalization layers. Furthermore, the residual mapping uses a 1×1 convolution layer, a batch normalization layer, and an element-wise addition to operate on the input; the detailed architecture of the residual module is depicted in Figure 3.

#### 3.2.2. Residual Pooling Layer

Mao [27] proposed a deep residual pooling network for texture recognition. Notably, the dimension of the feature vector generated by this network based on the texture image data is lower than that of previous deep texture recognition methods; however, it demonstrated state-of-the-art performance on benchmark texture recognition datasets. The residual pooling layer of the deep residual pooling network can be easily integrated with any CNN architecture to construct an end-to-end learning network that consists of a residual encoding module and an aggregation module.

Residual Encoding Module

The architecture of the residual pooling layer is depicted in Figure 4a, which is composed of a convolution transfer module and a short connection module, represented by green and red rectangles, respectively. The convolution transfer module consists of a convolutional layer, dropout layer, batch normalization layer, and a sigmoid layer as the activation function. Each of these layers has a size of and a stride of one. Dropout and batch normalization layers are used to avoid overfitting, and a sigmoid function is used to accentuate the difference between learned features and features learned from the preceding residual module. The short connection module is a sigmoid function whose output characteristics are added to those of the convolution transfer module, which forces the convolutional transfer module to learn to extract more iris texture features and accelerates the training process. Using the residual encoding module, more iris texture features can be extracted to improve the accuracy of iris verification.

b.Aggregation Module

The architecture of the aggregation module, which consists of a batch normalization layer preceded and followed by a ReLU layer, is depicted in Figure 4b. A global average pooling layer is used to compute the feature map to obtain the feature vector. The aggregation module yields the final feature vector with a dimension equal to the number of categories of the iris; no separate dimension reduction step is involved, which helps avoid the potential risk of overfitting and additional calculations.

### 3.3. Datasets

The following four databases were used for the experiments: (1) CASIA Iris Image Database V4-Distance (CASIA.v4-distance) [33], (2) ND Cross-Sensor Iris 2013 Dataset-LG4000 (ND-LG4000) [34], (3) CASIA Iris Image Database V4-Lamp (CASIA.v4-lamp) [33], and (4) CASIA Iris Image Database M1-S2 (CASIA-M1-S2) [33]. During the iris localization stage, we randomly selected 100 iris images from the 4 databases for the corresponding experimental study. During the iris verification stage, compared with other image datasets, such as ImageNet, the four iris datasets used in this paper are smaller. In order to prevent over fitting in the training process, the samples in each database were divided into 3 subsets: 80% for training, 10% for verification, and 10% for testing. Based on this rule, datasets are divided to make the training set as large as possible. Specifically, the following four publicly available iris databases were used in the experiments:(1)CASIA.v4-distance: CASIA.v4-distance contains 2567 iris images from 142 subjects. Iris images were collected by the Chinese Academy of Sciences Institute of Automation (CASIA) using a CASIA long-range iris camera. The test set for the performance evaluation consists of 130,321 match scores.(2)ND-LG4000: ND-LG4000 contains 29,986 iris images acquired from 1352 subjects using an LG 4000 iris biometrics sensor. The test set for the performance evaluation consists of 8,810,866 match scores.(3)CASIA.v4-lamp: CASIA.v4-lamp contains 16,212 iris images from 819 subjects. This dataset was acquired by turning a lamp on/off near the subject to produce elastic deformation under different lighting conditions. The test set for the performance evaluation consists of 2,624,400 match scores.(4)CASIA-M1-S2: CASIA-M1-S2 contains 6000 iris images from 400 subjects. Images of this dataset are acquired from mobile devices. The test set for the performance evaluation consists of 1,440,000 match scores.

## 4. Experimental Results and Discussion

### 4.1. Experimental Environment

The iris localization stage was implemented in C++ using Opencv. The iris recognition stage was implemented in Pytorch, and we trained the proposed iris feature extraction network architecture using a stochastic gradient descent optimizer with an initial learning rate of 0.001, momentum of 0.9, and weight decay of 0.0001. All iris localization and iris verification stages were equipped with an AMD-3900x CPU and NVIDIA 3090 GPU.

### 4.2. Discussion and Result Analysis

#### 4.2.1. Investigation of Iris Localization

Here, we present an analysis of the performance of the proposed iris localization method obtained by testing 400 iris images randomly selected from the 4 databases. Simultaneously, to reflect the advantages of the process acceleration induced by the GPU, we conducted iris localization experiments under equivalent conditions on both the GPU and CPU.

Table 1 presents a comparison between the average running times of the serial and parallel algorithm for the iris localization stage. The implementation of the parallel algorithm of the proposed scheme on various databases reveals that the average speed-up on the CASIA.v4-distance, ND-LG4000, CASIA.v4-lamp, and CASIA-M1-S2 datasets is 26, 32, 36, and 21 times, respectively, indicating that the flexibility of the CUDA in performing parallel alignment accelerates the iris localization step. In actual experiments, circular Hough transform takes up most of the running time of the whole iris location, reaching more than 95%. Therefore, reducing the running time of circular Hough transform can effectively reduce the running time of the whole iris positioning. The experimental results confirm that the parallel algorithm of this scheme can effectively reduce the running time of the whole iris location.

The accuracy of iris localization is also an important indicator. Table 2 presents the accuracy of iris localization achieved using this method. The accuracy of iris location is the percentage of the number of correctly located irises to the total number of located irises. As shown in Table 2, the proposed method achieved good results on the 4 iris datasets collected in a non-cooperative environment; the 4 different iris datasets all have a localization accuracy of more than 80%. To better comprehend the iris localization results, we selected certain challenging samples and presented the localization results obtained on the CASIA.v4-distance, ND-LG4000, CASIA.v4-lamp, and CASIA-M1-S2 datasets (Figure 5). In Figure 5, the inner and outer boundaries of the iris are marked by white circles; it is obvious that the iris location algorithm accurately extracts the inner and outer boundaries of the iris. We believe that this is due to the use of the iris location scheme proposed in this paper. The circular Hough transform and integro-differential operator are the mainstream iris inner and outer boundary location algorithms. The experimental results effectively confirm the advantages of these two algorithms.

#### 4.2.2. Evaluation of the Iris Verification Performance

To evaluate various approaches quantitatively, detection error trade-off (DET) curves were drawn. The equal error rate (EER) and false rejection rate (FRR), when the false accept rate (FAR) was $10^{-4}$, were used as measures.

The proposed iris verification method was evaluated on each database. To assess the effect of the residual pooling layer, several ablation experiments were conducted by ablating the residual pooling layer in the proposed network. The comparison results between iris recognition with and without the residual pooling layer are presented in Table 3 and Figure 6, respectively. As shown in Figure 6, on four datasets, with the same FAR, the FRR size of the proposed model is smaller than ResNet, indicating that the residual pooling layer has obvious advantages in improving iris verification accuracy. As shown in Table 3, on 4 datasets, the proposed network architecture outperforms 10.83%, 61.41%, 49.78%, and 36.65% of FRR when the FAR was $10^{-4}$, respectively, and 57.31%, 49.53%, 61.14%, and 52.36% of ERR, respectively, compared to the network architecture without the residual pooling layer.

As mentioned in the Materials and Methods section, a residual pooling layer is introduced into the proposed deep learning architecture. The residual pooling layer has the short connection module, the convolutional layer, and the sigmoid function, which make the iris texture feature extraction more effective. Therefore, the corresponding ablation experimental results show that adding a residual pooling layer can obtain higher iris recognition accuracy. On the other hand, compared with ResNet, the additional residual pooling layer increases the complexity of the deep learning model, which is the disadvantage of the deep learning architecture proposed in this paper.

To ascertain the effectiveness of the proposed network architecture in iris feature extraction, several highly competitive lightweight networks were used as benchmarks. Three typical deep learning networks, ResNet-18 [26], MobileNet-v2 [35], and Maxout-CNNs [28], were selected for the experimental comparison.

The comparative experimental results obtained using the proposed method and the respective benchmark methods are depicted in Figure 7. The EER and FRR, when the FAR was $10^{-4}$, obtained from the respective methods, are summarized in Table 4. As shown in Figure 6, on four datasets, with the same FAR, the FRR size of the proposed model is smaller than ResNet-18, MobileNet-v2, and Maxout-CNNs, respectively, indicating that the proposed model has obvious advantages in improving iris verification accuracy.

Furthermore, the extent of performance improvement varies among the databases; the observed improvement is also supported by the EER and FRR@FAR = $10^{-4}$ in Table 4. The EER and FRR@FAR = $10^{-4}$ of the proposed model are smaller than the other methods on the four iris datasets.

Maxout CNNs is a classic lightweight iris recognition deep learning architecture. Experiments on four datasets show that the iris recognition accuracy of the proposed depth learning architecture is higher than Maxout CNNs. We believe that this is because the proposed method uses the residual pooling layer and ResNet. More importantly, the network depth difference between Maxout CNNs and the proposed deep learning architecture is small, and we have selected the same super parameters for training, so, in fact, the recognition accuracy beyond Maxout CNNs is not very obvious. ResNet-18 is an architecture widely used in various fields of image processing. The biggest difference between ResNet-18 and the proposed method is the lack of a residual pooling layer, and the network depth is much higher than the proposed method. A too deep network is not conducive to iris recognition, because the number of iris datasets is relatively small compared with other large datasets, such as ImageNet. Therefore, the iris recognition accuracy of the method proposed in this paper is higher than ResNet-18. As an effective lightweight classical depth learning architecture, MobileNet-v2 also performs well in iris recognition accuracy, which is only a little lower than the proposed method.

Figure 8 illustrates the confusion matrices for iris verification obtained from the experimental results on the datasets of CASIA-Distance, ND-LG4000, CASIA-Lamp, and CASIA-M1-S2, respectively. In the four iris datasets, all irises verified True Negative (TN) reached 99% (green grid in the upper left corner). Since the number of genuine match scores is far less than the number of imposter match scores, the result of True Positive (TP) is lower than TN, but still higher than False Negative (FN) and False Positive (FP). Such high TN and TP results indicate that the proposed model realizes high-precision iris verification.

Table 5 provides the number of parameters and float point operations (FLOPs) from the proposed network architecture and other network architectures for comparison. Our machine configuration is AMD 3900X with 32 GB memory, and all the experiments use one NVIDIA RTX 3090 card with 24GB memory. The actual number of parameters and FLOPs will be different for different devices and input sizes, so the indicators in Table 5 are not fixed. We input a 128 × 128 image into each network architecture to obtain the number of parameters and FLOPs. As shown in Table 5, the number of network architecture parameters proposed in this paper is smaller than those used for comparison. In terms of FLOPs, the network architectures proposed in this paper are smaller than those used for comparison but larger than MobileNet-v2, and this is a defect of the proposed model in this paper.

## 5. Conclusions

In this study, we developed an efficient iris recognition system consisting of an iris localization and iris verification stage. To solve the problem of fast iris localization, we used a parallel circular transform to locate the inner boundary of the iris, after which the fast Daugman integro-differential localization algorithm was used to achieve effective and rapid iris localization. Furthermore, we proposed a lightweight deep learning architecture with a residual pooling layer to achieve high-precision iris recognition with less model parameters. The proposed iris localization method was implemented using 400 iris images obtained from four different representative iris databases, and the speed and effectiveness of this method were verified. The proposed iris verification method was evaluated using four representative iris image databases, and the performance of the deep learning architecture for high-precision iris recognition was verified. In general, the system can effectively realize iris recognition. First, the proposed positioning method was used to achieve a fast and accurate iris localization stage, and then the proposed deep learning architecture was used to complete high-precision iris verification.

Although the proposed iris recognition scheme has the many advantages above, it also has some defects. As there are two different schemes, the classical manual feature engineering method and the deep learning method, the problem is how to deploy the entire iris recognition process to the embedded device. Another problem is that the iris images we deal with are all illuminated under near infrared light. The iris localization scheme proposed in the paper can effectively extract the inner and outer boundaries of the iris, but for the irregular iris regions formed under visible light illumination, this scheme cannot achieve accurate iris localization. As for future work, we have considered deploying the proposed system to the actual embedded architecture to achieve real-time iris recognition and optimizing the iris positioning algorithm to achieve iris accurate localization under visible lighting conditions.

## Figures and Tables

**Figure 1 sensors-22-07723-f001:**
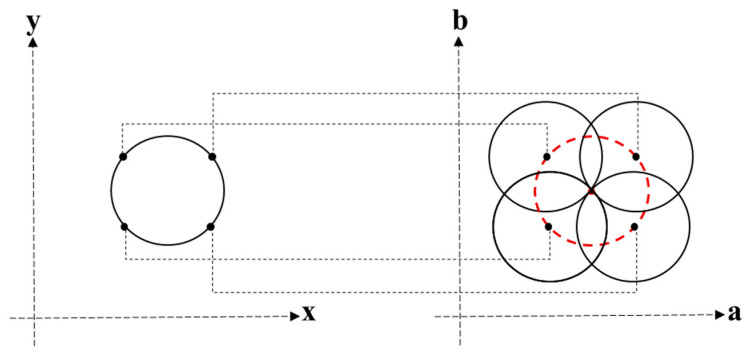
Conversion of the image space into the Hough space.

**Figure 2 sensors-22-07723-f002:**
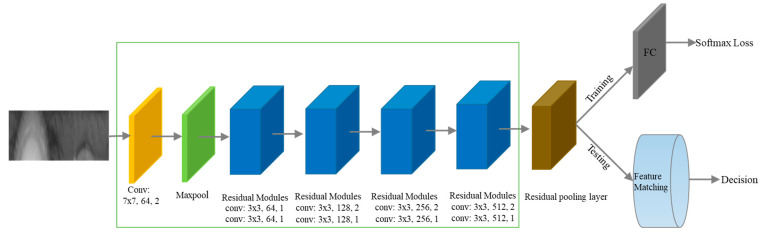
Architecture of the proposed network for iris verification (convolutional layer configuration denoted as kernel size, number of channels, and stride). A CNN integrated with a residual pooling layer is developed for extracting more texture features of the iris. The softmax loss function is used for training.

**Figure 3 sensors-22-07723-f003:**
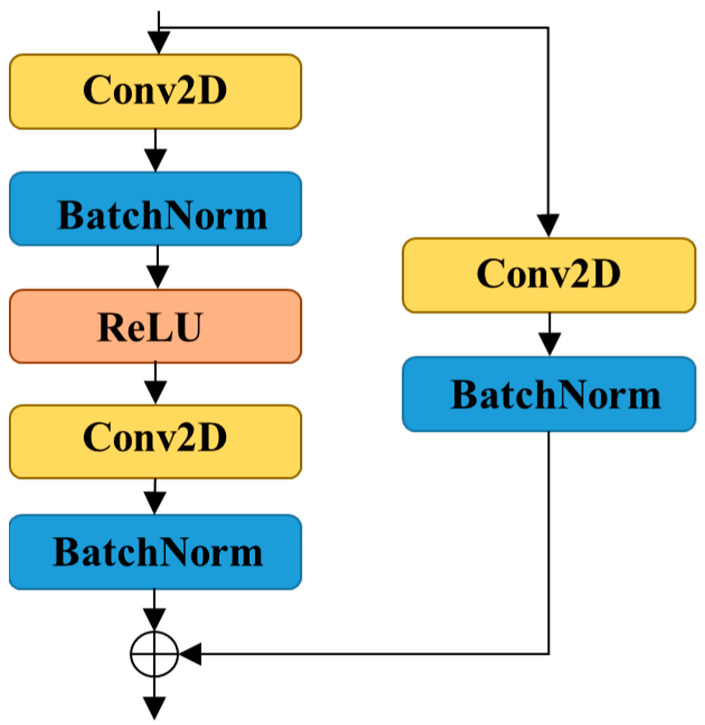
Illustration of the residual module.

**Figure 4 sensors-22-07723-f004:**
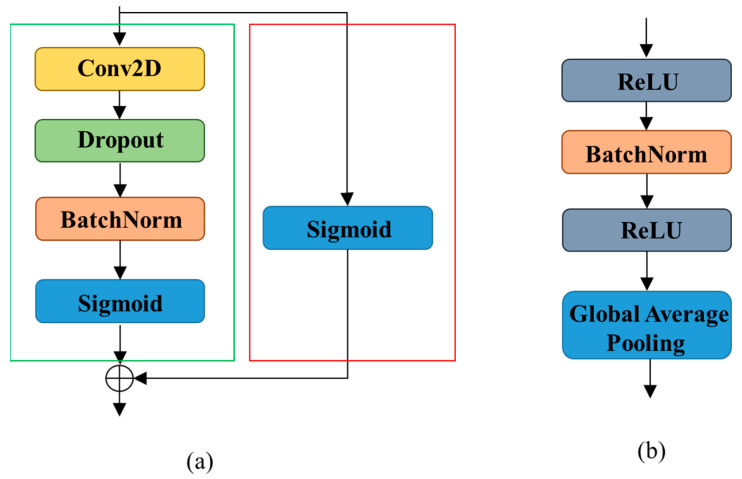
Two components of the proposed residual pooling layer. (**a**) Residual encoding module and (**b**) aggregation module.

**Figure 5 sensors-22-07723-f005:**
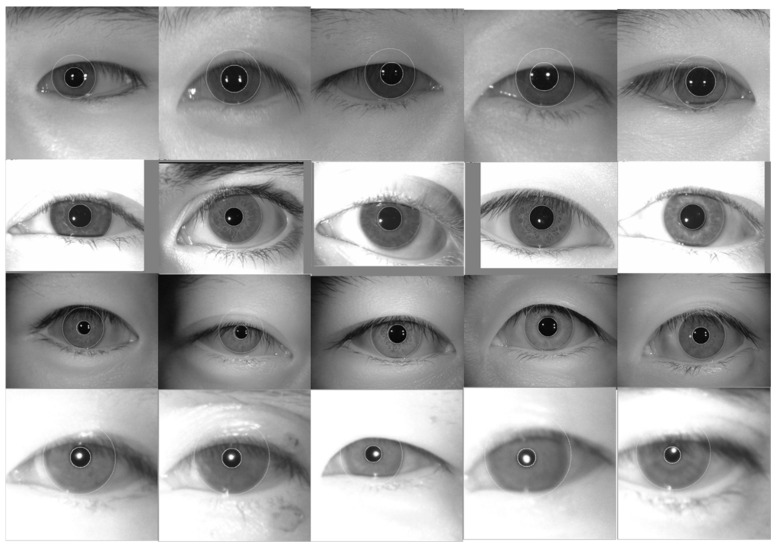
Typical iris localization results obtained using the proposed method on the CASIA.v4-distance, ND-LG4000, CASIA.v4-lamp, and CASIA-M1-S2 datasets.

**Figure 6 sensors-22-07723-f006:**
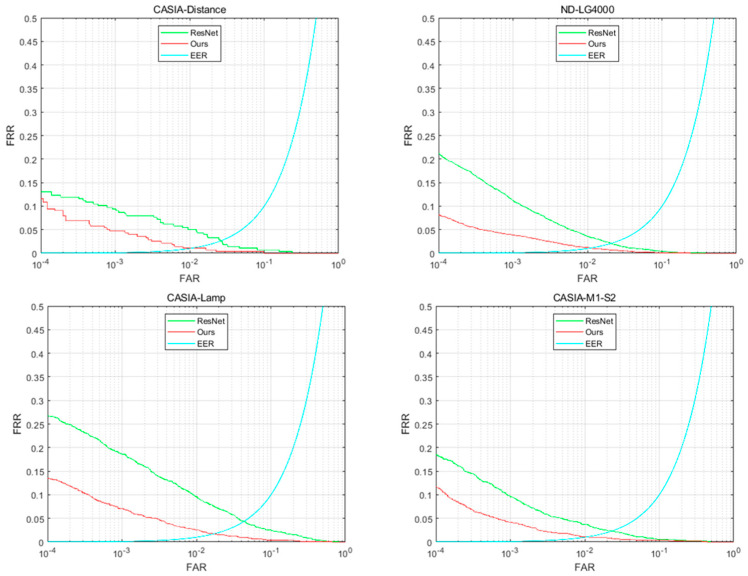
DET curves obtained from iris verification experiments conducted using ResNet and the proposed method.

**Figure 7 sensors-22-07723-f007:**
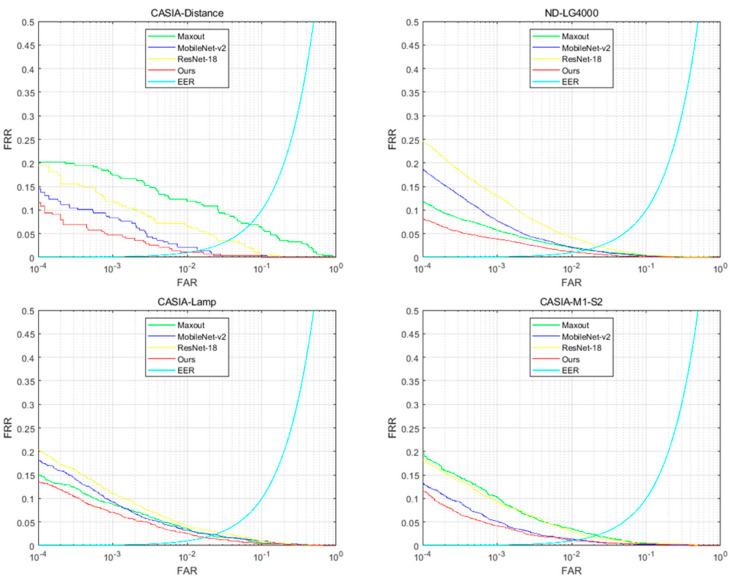
DET curves obtained from iris verification experiments conducted using ResNet-18, Maxout-CNNs, MobileNet-v2, and the proposed method.

**Figure 8 sensors-22-07723-f008:**
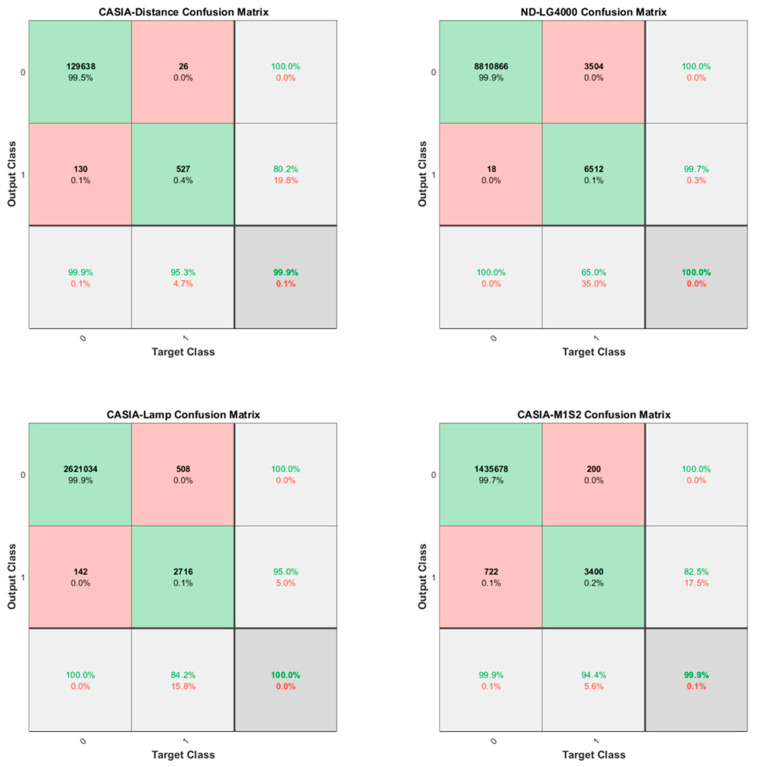
Obtained confusion matrix of the proposed model for iris recognition.

**Table 1 sensors-22-07723-t001:** Comparison between the average runtime of the serial and parallel algorithms at the iris localization stage on different databases.

Database	Tp(S)	Ts(S)	Speedup
CASIA.v4-distance	20.63	0.79	26x
ND-LG4000	41.26	1.16	36x
CASIA.v4-lamp	23.13	0.73	32x
CASIA-M1-S2	13.31	0.64	21x

**Table 2 sensors-22-07723-t002:** Accuracy of iris localization.

Database	Localization Accuracy
CASIA.v4-distance	94%
ND-LG4000	90%
CASIA.v4-lamp	91%
CASIA-M1-S2	85%

**Table 3 sensors-22-07723-t003:** Performance of the proposed network and ResNet.

Recognition Approach	CASIA-Distance		ND-LG4000		CASIA-Lamp		CASIA-M1-S2	
	FRR@FAR = 0.01%	EER	FRR@FAR = 0.01%	EER	FRR@FAR = 0.01%	EER	FRR@FAR = 0.01%	EER
ResNet	13.38%	2.53%	21.17%	2.12%	26.80%	4.40%	18.50%	2.33%
Ours	11.93%	1.08%	8.17%	1.07%	13.46%	1.71%	11.72%	1.11%

**Table 4 sensors-22-07723-t004:** Comparisons between different iris verification networks.

Recognition Approach	CASIA-Distance		ND-LG4000		CASIA-Lamp		CASIA-M1-S2	
	FRR@FAR = 0.01%	EER	FRR@FAR = 0.01%	EER	FRR@FAR = 0.01%	EER	FRR@FAR = 0.01%	EER
ResNet-18	19.89%	3.61%	25.78%	2.33%	21.53%	2.73%	18.11%	1.77%
MobileNet-v2	14.83%	1.44%	20.07%	1.65%	19.85%	2.36%	13.28%	1.18%
Maxout-CNNs	20.25%	6.88%	12.48%	1.58%	15.51%	2.11%	19.72%	2.18%
Ours	11.93%	1.08%	8.17%	1.07%	13.46%	1.71%	11.72%	1.11%

**Table 5 sensors-22-07723-t005:** Comparison of the number of parameters in the model architectures.

Recognition Approach.	#Params	FLOPs
ResNet-18	14.53 MB	2.24 G
MobileNet-v2	11.78 MB	111.91 M
Maxout-CNNs	10.12 MB	1.27 G
Ours	8.79 MB	274.48 M

## Data Availability

No applicable.

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
