# Peer review of "Iris Recognition Method Based on Parallel Iris Localization Algorithm and Deep Learning Iris Verification"

_sensors, 2022, doi:10.3390/s22207723_

Round 1
Reviewer 1 Report
The title should be reframed to reflect the actual representation of the research.
The Abstract is well written, however, in line 13 the author started a sentence with 'Because'. Also, the result section of the abstract has more quantitative results than the only reported one.
Keyword: The author should include the specific CCN architecture used in the keyword.
Section 4.1 Datasets should be in Section 3. Material and Methods as datasets are part material. If you have curated the dataset yourself then it would be part of your result sections. Also, the datasets are not well described. No reason was given for using 80:10:10 for your experiments.
Section 4.3.1: In a logical manner according to writing Table 1 and Table 2 should come first before Figure 5. Similarly, line313 & 314 revise Table 3 and Figure 6 to reflect the order of presentation.
Author Response
Response to Reviewer 1 Comments
Point 1, The title should be reframed to reflect the actual representation of the research.
Response 1: Thanks for your kind comments. This is a good idea. We have designed a new title “Iris Recognition Method Based on Parallel Iris Localization Algorithm and Deep Learning Iris verification”.
Point 2, The Abstract is well written, however, in line 13 the author started a sentence with 'Because'. Also, the result section of the abstract has more quantitative results than the only reported one.
Response 2: We rewrote the Abstract section and added more quantitative results, as shown below:
Biometric recognition technology has been widely used in various fields of society. Iris recognition technology, as a stable and convenient biometric recognition technology, has been widely used in security applications. However, the iris images collected in the actual non cooperative environment have various noises. Although the mainstream iris recognition methods based on deep learning have achieved good recognition accuracy, the condition is to increase the complexity of the model. On the other hand, what the actual optical system collects is the original iris image that is not normalized. The mainstream iris recognition scheme based on deep learning does not consider the iris localization stage. In order to solve the above problems, this paper proposes an effective iris recognition scheme consist of iris localization and iris verification stage. For the iris localization stage, we used the parallel Hough circle to extract the inner circle of the iris and the Daugman algorithm to extract the outer circle of the iris, and for the iris verification stage, we developed a new lightweight convolutional neural network. The architecture consists of a ResNet module, and a residual pooling layer which is introduced to effectively improve the accuracy of iris verification. Iris localization experiments were conducted on 400 iris images collected under a non-cooperative environment. Compared with its processing time on a graphics processing unit with a compute processing unified architecture, the experimental results revealed that the speed was increased by 26, 32, 36 and 21 times at four different iris datasets, respectively, and the effective iris localization accuracy is achieved. Furthermore, we choose four representative iris datasets collected under a non-cooperative environment for the iris verification experiments. The experimental results demonstrated that the network structure could achieve high-precision iris verification with fewer parameters, and the equal error rates are 1.08%, 1.01%, 1.71% and 1.11% on four test databases, respectively.
Point 3, Keyword: The author should include the specific CCN architecture used in the keyword.
Response 3: We added the feature CNN to the keyword, as shown below.
Keywords: Iris recognition, iris localization, iris verification, deep residual network, residual pooling layer.
Point 4, Section 4.1 Datasets should be in Section 3. Material and Methods as datasets are part material. If you have curated the dataset yourself then it would be part of your result sections. Also, the datasets are not well described. No reason was given for using 80:10:10 for your experiments.
Response 4: Thanks for your professional comments, We have placed Section 4.1 in Section 3 and rewritten it, as shown below:
The following four databases were used for the experiments: (1) CASIA Iris Image Database V4-Distance (CASIA.v4-distance), (2) ND Cross-Sensor Iris 2013 Dataset-LG4000 (ND-LG4000), (3) CASIA Iris Image Database V4-Lamp (CASIA.v4-lamp), and (4) CASIA Iris Image Database M1-S2 (CASIA-M1-S2). During the iris localization stage, we randomly selected 100 iris images from the four databases for the corresponding experimental study. During the iris verification stage, compared with other image datasets, such as ImageNet, the four iris datasets used in this paper are smaller. In order to prevent over fitting in the training process, the samples in each database were divided into three subsets: 80% for training, 10% for verification, and 10% for testing. Based on this rule, datasets are divided to make the training set as large as possible. Specifically, the following four publicly available iris databases were used in the experiments:
CASIA.v4-distance: CASIA.v4-distance contains 2567 iris images from 142 subjects. Iris images were collected by the Chinese Academy of Sciences Institute of Automation (CASIA) using a CASIA long-range iris camera. The test set for the performance evaluation consists of number of 130321 match scores.
ND-LG4000: ND-LG4000 contains 29986 iris images acquired from 1352 subjects using LG 4000 iris biometrics sensor. The test set for the performance evaluation consists of number of 8810866 match scores.
CASIA.v4-lamp: CASIA.v4-lamp contains 16212 iris images from 819 subjects. This dataset was acquired by turning on/off a lamp near the subject to produce elastic deformation under different lighting conditions. The test set for the performance evaluation consists of number of 2624400 match scores.
CASIA-M1-S2: CASIA-M1-S2 contains 6000 iris images from 400 subjects. Images of this dataset are acquired from mobile devices. The test set for the performance evaluation consists of number of 1440000 match scores.
Point 5, Section 4.3.1: In a logical manner according to writing Table 1 and Table 2 should come first before Figure 5. Similarly, line313 & 314 revise Table 3 and Figure 6 to reflect the order of presentation.
Response 5: We have completed the revise of corresponding tables and figures in the manuscript. The Table 1 and Table 2 come first before Figure 5, Table 3 and Figure 6, Table 4 and Figure 7 are interchanged respectively.

Reviewer 2 Report
Decision: Extensive revision.
Comments: The idea presented in this paper is worthy and can be proved beneficial in terms of an effective approach to recognition. The presentation of the paper is not convinced there are some major and minor flaws, comments, and suggestions for improvement in this paper which should be addressed for possible acceptance in this journal. looking forward to seeing these changes in the revised version.
· I have concerns about the proposed methods, employ which methodologies? Which problem still requires to be solved?
· Why is the proposed approach suitable to be used to solve the critical problem? We need a more convincing response to indicate clearly the SOTA development.
· What is the main difficulty when applying the proposed method? The authors should clearly state the limitations of the proposed method in practical applications and should be mentioned in the article's conclusion.
· Please discuss the hyperparameters setting of the proposed model and other comparable models.
· Please discuss your findings and what the advantage of your proposed model is.
· “Discussion” section should be added in a more highlighting, argumentative way. The author should analyze the reason why the test results are achieved.
· The complexity of the proposed model and the model parameter uncertainty is not mentioned.
· The abstract should give the reader a quick, concise, and very brief summary of the paper’s content. In this paper, the abstract portion is not properly written and is very short. The actual aim of the abstract is unfortunately not described properly. It is suggested that the abstract structure should be rewritten according to the following structure:
· The first two or three sentences should be according to the subject and background. And then further suggested that the structure should be like Problem>> Aim>> Method>> Results and findings.
· The proposed solution to the problem given in the introduction section is not clearly explained. It needs proper attention of the authors and states the limitations of existing systems and clear solutions of your system to highlight the main influence of this work on over-employed techniques.
§ The authors should follow the structure of the research article and provide background knowledge of the domain.
§ Please avoid the basic grammatical mistakes.
§ The redundant words should be replaced with proper alternative relevant words.
§ Please provide proper existing research and challenges, followed by the contribution of the researchers to handle those challenges.
§ I suggest that the authors should write clear contributions which fill the gap in the existing work.
· Overall, Tables, and Figures should be explained comprehensively.
· The proper graph should be included for verifying the claimed performance such as the confusion matrix which is very necessary.
· The results of real-time recognition of Chinese speakers should be included to accurately verify the testing performance of the proposed framework.
· The conclusion of this research is very weak, the authors have not summarized the overall research work.
· References are not enough and few, do not contradict and cover a specific era more literature should be added. References to newer or more recent work are required. But if authors pass this round, it is more meaningful to provide next time.
· More recently published papers in the field of deep learning should be discussed in the Introduction/literature. The authors may be benefited by reviewing more papers such as: 10.3390/s21175892.
Author Response
Response to Reviewer 2 Comments
Point 1, I have concerns about the proposed methods, employ which methodologies? Which problem still requires to be solved?
Response 1: The method used in this manuscript is as follows: The iris recognition process comprises the following steps: image acquisition, iris localization, normalization, feature extraction, and matching. For iris localization steps, we used the parallel Hough circle to extract the inner circle of the iris and the Daugman algorithm to extract the outer circle of the iris. For the normalization step, we use the rubber sheet model. The normalized iris image can be obtained through iris localization and normalization steps. The above steps belong to the classical manual feature engineering method. The steps of iris feature extraction and matching are completed by convolutional neural network. The convolutional neural network architecture proposed in the manuscript is used to achieve iris feature extraction, and then the cosine similarity measurement is implemented for the iris feature vectors output by different kinds of iris through the network architecture, and the matching score is obtained as the result of iris recognition. These steps belong to the deep learning-based methods.
The problems to be solved are as follows: The current problems to be solved are as follow: At present, the mainstream iris recognition method based on deep learning has the problem of too many model parameters, which makes it difficult to deploy the model to embedded devices. This manuscript proposes a new lightweight deep learning model, and adds a residual pooling layer to improve the accuracy of iris recognition to solve the problem. Another problem is that the mainstream iris recognition method based on deep learning takes the normalized iris image as the input, and the actual optical system collects the original eye image. In this paper, we consider the actual iris recognition system, and propose a scheme to quickly extract the inner and outer boundaries of the iris from eye images. Then, through the normalization step, the required normalized iris image can be obtained as the input of the subsequent neural network architecture.
Point 2, Why is the proposed approach suitable to be used to solve the critical problem? We need a more convincing response to indicate clearly the SOTA development.
Response 2: we choose several representative papers that use deep learning methods to achieve high-precise iris recognition. Liu [1] tried to solve the problem of heterogeneous iris recognition using deep learning method, which named “Deepiris”. Unlike the manual feature extraction and feature matching in traditional solutions, DeepIris directly learns the nonlinear mapping function between a pair of iris images, and uses paired filter banks from different sources for identity monitoring. On the other hand, DeepIris learns a pair of filter banks to establish the relationship between heterogeneous iris images. Compared with the traditional manual iris recognition scheme, DeepIris achieves the most advanced performance.
Gangwar A [2] proposed DeeplrisNet architecture for iris recognition in the early stage. This scheme is mainly aimed at iris recognition under non cooperative conditions. It is based on very deep architecture and various skills from the latest successful CNNs. The experimental results show that the architecture can provide robustness, discriminability, compactness and very easy to implement iris representation to obtain the most advanced accuracy. The problem is that the parameters of the deep convolution neural network are too large and the model complexity is high.
Zhao [3] proposed a full convolutional network based deep learning architecture to implement iris recognition, named UniNet. Using this architecture, spatial corresponding iris feature descriptors are generated, and a specially designed extended triple loss function is used to learn and identify iris spatial features. Using this architecture, advanced iris recognition accuracy can be achieved. Based on the above deep learning iris recognition method, Wang [4] proposed DRFNet to achieve more accurate iris recognition. Unlike UniNet, DRFNet introduces residual network learning with extended convolution kernel to optimize the learning process and aggregate context information from iris images. The iris recognition accuracy of this algorithm is higher than that of UniNet.
Iris and periocular fusion recognition has also developed rapidly in recent years. Zhang [5] proposed a deep feature fusion network, which uses the complementary information of iris and periocular region to improve the performance of mobile recognition. Firstly, the convolution neural network model with the maxout unit is used to extract the robust, compact and discriminative features of iris and periocular region. then, the iris and periocular depth features are fused by weighted stitching. The experiment on near infrared moving iris database shows that the convolution neural network model used in this method is effective, and the deep feature fusion method based on adaptive weights has achieved good results. Wang [6] proposed a new low constraint iris recognition framework, which has tried to use better eye circumference matching and introduce iris recognition with more accurate similarity score. The experimental results prove the advantages of the proposed method and obtain excellent results.
[1] N. Liu, M. Zhang, H. Li, Z. Sun, and T. Tan, “DeepIris: Learning pairwise filter bank for heterogeneous iris verification,” Pattern Recognit. Lett., vol. 82, pp. 154–161, 2016
[2] A. Gangwar and A. Joshi, “DeepIrisNet: Deep iris representation with applications in iris recognition and cross-sensor iris recognition”, in Proc. IEEE Int. Conf. Image Process. (ICIP), Sep. 2016, pp. 2301–2305
[3] Z. Zhao and A. Kumar, “Towards more accurate iris recognition using deeply learned spatially corresponding features,” in Proc. IEEE Int. Conf. Comput. Vis., Venice, Italy, Oct. 2017, pp. 3829–3838.
[4] K. Wang and A. Kumar, “Towards more accurate iris recognition using dilated residual features,” IEEE Trans. Inf. Forensics Security, vol. 14, no. 12, pp. 3233–3245, Dec. 2019.
[5] Q. Zhang, H. Li, Z. Sun, and T. Tan, “Deep feature fusion for iris and periocular biomet-rics on mobile devices,” IEEE Trans. Inf. Forensics Security, vol. 13, no. 11, pp. 2897–2912, Nov. 2018.
[6] Wang K, Kumar A, “Periocular-assisted multi-feature collaboration for dynamic iris recognition,” IEEE Trans. Inf. Forensic Secur 16:866–879, 2020.
Point 3, What is the main difficulty when applying the proposed method? The authors should clearly state the limitations of the proposed method in practical applications and should be mentioned in the article's conclusion.
Response 3: We completed the above iris recognition process on the server. Because there are two different schemes, namely, the classical manual feature engineering method and the deep learning method, the problem is how to deploy the entire iris recognition process to the embedded device. Another problem is that the iris images we deal with are all illuminated under near infrared light. The iris localization scheme proposed in the manuscript can effectively extract the inner and outer boundaries of the iris, but for the irregular iris regions formed under visible light illumination, this scheme cannot achieve accurate iris localization.
We put the above contents into the conclusion section, as shown below:
Although the proposed iris recognition scheme has many advantages above, it also has some defects. As there are two different schemes, the classical manual feature engineering method and the deep learning method, the problem is how to deploy the entire iris recog-nition process to the embedded device. Another problem is that the iris images we deal with are all illuminated under near infrared light. The iris localization scheme proposed in the paper can effectively extract the inner and outer boundaries of the iris, but for the ir-regular iris regions formed under visible light illumination, this scheme cannot achieve accurate iris localization.
Point 4, Please discuss the hyperparameters setting of the proposed model and other comparable models.
Response 4: ResNet is tested and compared with the proposed method to evaluate the effectiveness of residual pooling layer, which has the same hyperparameters as the proposed deep learning model by using a stochastic gradient descent optimizer with an initial learning rate of 0.001, batch size of 64, momentum of 0.9, and weight decay of 0.0001.
Three typical deep learning networks, ResNet-18, MobileNet-v2, and Maxout-CNNs were selected for the experimental comparison.In order to demonstrate the excellent performance of the deep learning architecture proposed in this manuscript, we train these three typical deep learning architectures to use the same hyperparameters as the proposed method by using a stochastic gradient descent optimizer with an initial learning rate of 0.001, batch size of 64, momentum of 0.9, and weight decay of 0.0001.
Point 5, Please discuss your findings and what the advantage of your proposed model is.
Response 5: He proposed ResNet for image recognition. This architecture can effectively increase accuracy and solve the problem of over fitting, which has great advantages for training the smaller iris datasets. Mao introduced a residual pooling layer for texture recognition, which can preserve the spatial information of texture to achieve advanced recognition performance. Iris has obvious texture features, which motivates us to use this architecture for iris recognition. Based on the above, this paper proposes a deep learning architecture combining ResNet and residual pooling layer for iris verification. The new lightweight deep learning architecture for iris verification composed of several residual modules. A simple residual pooling layer was added to extract more texture features of the iris to improve the accuracy of iris recognition. With this network architecture, high-precision iris verification could be realized with low model complexity. More importantly, considering that the actual iris recognition optical system captures the original iris image that is not normalized, we used the parallel circular Hough transform to locate the inner boundary of the iris, after which the fast Daugman integro-differential localization algorithm was used to locate the outer boundary based on the inner boundary information. Thus, a fast and effective iris localization scheme in a non-cooperative environment was realized. Then the normalization step can be realized by using the inner and outer boundary information of the iris, and the input of the iris verification deep learning architecture can be obtained.
The above content has been added in the Introduction section.
Point 6, “Discussion” section should be added in a more highlighting, argumentative way. The author should analyze the reason why the test results are achieved.
Response 6: we improved the discussion section in the manuscript, as shown below:
Here, we present an analysis of the performance of the proposed iris localization method obtained by testing 400 iris images randomly selected from the four databases. Simultaneously, to reflect the advantages of the process acceleration induced by the GPU, we conducted iris localization experiments under equivalent conditions on both the GPU and CPU.
Table 1 presents a comparison between the average running times of the serial and parallel algorithm for the iris localization stage. The implementation of the parallel algorithm of the proposed scheme on various databases reveals that the average speed-up on the CASIA.v4-distance, ND-LG4000, CASIA.v4-lamp, and CASIA-M1-S2 datasets is 26, 32, 36 and 21 times, respectively, indicating that the flexibility of the CUDA in performing parallel alignment accelerates the iris localization step. In actual experiments, circular Hough transform takes up most of the running time of the whole iris location, reaching more than 95%. Therefore, reducing the running time of circular Hough transform can effectively reduce the running time of the whole iris positioning. The experimental results confirm that the parallel algorithm of this scheme can effectively reduce the running time of the whole iris location.
The accuracy of iris localization is also an important indicator. Table 2 presents the accuracy of iris localization achieved using this method. The accuracy of iris location is the percentage of the number of correctly located iris to the total number of located iris. As shown in Table 2, the proposed method achieved good results on the four iris datasets collected in a non-cooperative environment, four different iris datasets all have a localization accuracy of more than 80%. To better comprehend the iris localization results, we selected certain challenging samples and presented the localization results obtained on the CASIA.v4-distance, ND-LG4000, CASIA.v4-lamp, and CASIA-M1-S2 datasets (Fig. 5). In Fig. 5, the inner and outer boundaries of the iris are marked by white circles, it is obvious that the iris location algorithm accurately extracts the inner and outer boundaries of the iris. We believe that this is due to the use of the iris location scheme proposed in this paper. The circular Hough transform and integro-differential operator are the mainstream iris inner and outer boundary location algorithms. The experimental results effectively confirm the advantages of these two algorithms.
The proposed iris verification method was evaluated on each database. To assess the effect of the residual pooling layer, several ablation experiments were conducted by ablating the residual pooling layer in the proposed network. The comparison results between iris recognition with and without the residual pooling layer are presented in Table 3 and Fig. 6, respectively. As shown in Fig. 6, on four datasets, with the same FAR, the FRR size of the proposed model is smaller than ResNet, indicating that the residual pooling layer has obvious advantages in improving iris verification accuracy. As shown in Table 3, on four datasets, the proposed network architecture outperforms 10.83%, 61.41%, 49.78% and 36.65% of FRR when the FAR was respectively, and 57.31%, 49.53%, 61.14% and 52.36% of ERR respectively, compared to the network architecture without the residual pooling layer.
As mentioned in the Materials and Methods section, a residual pooling layer is introduced into the proposed deep learning architecture. The residual pooling layer has the short connection module, the convolutional layer and the sigmoid function, which make the iris texture feature extraction more effective. Therefore, the corresponding ablation experimental results show that adding residual pooling layer can obtain higher iris recognition accuracy. On the other hand, compared with ResNet, the additional residual pooling layer increases the complexity of the deep learning model, which is the disadvantage of the deep learning architecture proposed in this paper.
The comparative experimental results obtained using the proposed method and the respective benchmark methods are depicted in Fig. 7. The EER and FRR, when the FAR was , obtained from the respective methods are summarized in Table 4. As shown in Fig. 6, on four datasets, with the same FAR, the FRR size of the proposed model is smaller than ResNet-18, MobileNet-v2 and Maxout-CNNs respectively, indicating that the proposed model has obvious advantages in improving iris verification accuracy.
Furthermore, the extent of performance improvement varies among the databases; the observed improvement is also supported by the EER and FRR@FAR= in the Table 4. The EER and FRR@FAR= of the proposed model is smaller than the other methods on the four iris datasets.
Maxout CNNs is a classic lightweight iris recognition deep learning architecture. Experiments on four datasets show that the iris recognition accuracy of the proposed depth learning architecture is higher than Maxout CNNs. We believe that this is because the proposed method uses the residual pooling layer and ResNet. More importantly, the network depth difference between Maxout CNNs and the proposed deep learning architecture is small, and we have selected the same super parameters for training, so in fact, the recognition accuracy beyond Maxout CNNs is not very obvious. ResNet-18 is an architecture widely used in various fields of image processing. The biggest difference between ResNet-18 and the proposed method is the lack of residual pooling layer, and the network depth is much higher than the proposed method. Too deep network is not conducive to iris recognition, because the number of iris data sets is relatively small compared with other large data sets, such as ImageNet. Therefore, the iris recognition accuracy of the method proposed in this paper is higher than ResNet-18. As an effective lightweight classical depth learning architecture, Mobile-v2 also performs well in iris recognition accuracy, which is only a little lower than the proposed method.
Point 7, The complexity of the proposed model and the model parameter uncertainty is not mentioned.
Response 7: we added the discussion on the complexity of the model and the uncertainty of the model parameters in the manuscript, as shown below:
Table 5 provides the number of parameters and float point operations (FLOPs) from the proposed network architecture and other network architectures for comparison. Our machine configuration is AMD 3900x with 32 GB memory, and all the experiments use one NVIDIA RTX 3090 card with 24GB memory. The actual number of parameters and FLOPs will be different for different devices and input sizes, so the indicators in Table 5 are not determined. We input a 128 * 128 image into each network architecture to obtain the number of parameters and FLOPs. As shown in Table 5, the number of network architecture parameters proposed in this paper is smaller than those used for comparison. In terms of FLOPs, the network architectures proposed in this paper are smaller than those used for comparison but larger than MobileNet-v2, and this is a defect of the proposed model in this paper.
Table 5. Comparison of the number of parameters in the model architectures
|
Recognition Approach |
#Params |
FLOPs |
|
ResNet-18 |
14.53MB |
2.24G |
|
MobileNet-v2 |
11.78MB |
111.91M |
|
Maxout-CNNs |
10.12MB |
1.27G |
|
Ours |
8.79MB |
274.48M |
Point 8, The abstract should give the reader a quick, concise, and very brief summary of the paper’s content. In this paper, the abstract portion is not properly written and is very short. The actual aim of the abstract is unfortunately not described properly. It is suggested that the abstract structure should be rewritten according to the following structure:
The first two or three sentences should be according to the subject and background. And then further suggested that the structure should be like Problem>> Aim>> Method>> Results and findings.
Response 8: we rewrote the abstract in the manuscript, as shown below:
Biometric recognition technology has been widely used in various fields of society. Iris recognition technology, as a stable and convenient biometric recognition technology, has been widely used in security applications. However, the iris images collected in the actual non cooperative environment have various noises. Although the mainstream iris recognition methods based on deep learning have achieved good recognition accuracy, the condition is to increase the complexity of the model. On the other hand, what the actual optical system collects is the original iris image that is not normalized. The mainstream iris recognition scheme based on deep learning does not consider the iris localization stage. In order to solve the above problems, this paper proposes an effective iris recognition scheme consist of iris localization and iris verification stage. For the iris localization stage, we used the parallel Hough circle to extract the inner circle of the iris and the Daugman algorithm to extract the outer circle of the iris, and for the iris verification stage, we developed a new lightweight convolutional neural network. The architecture consists of a deep residual network module, and a residual pooling layer which is introduced to effectively improve the accuracy of iris verification. Iris localization experiments were conducted on 400 iris images collected under a non-cooperative environment. Compared with its processing time on a graphics processing unit with a central processing unit architecture, the experimental results revealed that the speed was increased by 26, 32, 36 and 21 times at four different iris datasets, respectively, and the effective iris localization accuracy is achieved. Furthermore, we choose four representative iris datasets collected under a non-cooperative environment for the iris verification experiments. The experimental results demonstrated that the network structure could achieve high-precision iris verification with fewer parameters, and the equal error rates are 1.08%, 1.01%, 1.71% and 1.11% on four test databases, respectively.
Point 9, The proposed solution to the problem given in the introduction section is not clearly explained. It needs proper attention of the authors and states the limitations of existing systems and clear solutions of your system to highlight the main influence of this work on over-employed techniques.
- The authors should follow the structure of the research article and provide background knowledge of the domain.
- Please avoid the basic grammatical mistakes.
- The redundant words should be replaced with proper alternative relevant words.
- Please provide proper existing research and challenges, followed by the contribution of the researchers to handle those challenges.
- I suggest that the authors should write clear contributions which fill the gap in the existing work.
Response 9: we rewrote the introduction section in the manuscript, as shown below:
Biometrics methods, which identify people based on physiological or behavioral characteristics, have been widely used in various commercial, civil, and forensic applications. As a biometric technology for identification, iris recognition can be used in several crucial places such as border crossings, banks, private companies, institutes, and law enforcement agencies. The iris is the annular region of the human eye between the sclera and pupil, and the complex iris texture with unique information is suitable for personal identification. However, Iris images collected under non-cooperative environments often contain various forms of noise, including reflection points, motion blur, and occlusions of glasses, hair, and eyelashes. This presents a challenge for iris recognition.
Recently, deep learning techniques have been widely used in several research fields, such as photonics research, biological imaging, material science, and image super-resolution, all of which demonstrate the advantages of these techniques. Similarly, the strategy based on deep learning has achieved excellent performance in a series of biometric technology problems, such as emotion recognition, gait recognition, fingerprint recognition and voice signal recognition. This also urges researchers to use deep learning methods to overcome iris recognition problems in non-cooperative environments. Gangwar A proposed DeeplrisNet architecture for iris recognition in the early stage. This scheme is based on the deep architecture and various tricks for successful applications of CNNs, which could achieve outstanding iris recognition performance based on the deep architecture. Wang proposed a capsule network architecture for iris recognition. By docking different outputs of several classical neural network structures with the capsule architecture, the experimental results indicated the feasibility and stability of the approach. Also, Minaee S uses VGGNet to extract iris deep features to achieve high-precision recognition results. Nguyen studied several pre-trained CNN models, including AlexNet, VGGNet, InceptionNet, ResNet, and DenseNet, to extract off-the-shelf CNN features for accurate iris recognition. Although the above methods could achieve high-precision iris recognition results, they are all at the cost of increasing the complexity of the model, and there is little research on iris location stage.
He proposed deep residual network (ResNet) for image recognition. This architecture can effectively increase accuracy and solve the problem of over fitting, which has great advantages for training the smaller iris datasets. Mao introduced a residual pooling layer for texture recognition, which can preserve the spatial information of texture to achieve advanced recognition performance. Iris has obvious texture features, which motivates us to use this architecture for iris recognition. Based on the above, this paper proposes a deep learning architecture combining ResNet and residual pooling layer for iris verification. The new lightweight deep learning architecture for iris verification composed of several residual modules. A simple residual pooling layer was added to extract more texture features of the iris to improve the accuracy of iris recognition. With this network architecture, high-precision iris verification could be realized with low model complexity. More importantly, considering that the actual iris recognition optical system captures the original iris image that is not normalized, we used the parallel circular Hough transform to locate the inner boundary of the iris, after which the fast Daugman integro-differential localization algorithm was used to locate the outer boundary based on the inner boundary information. Thus, a fast and effective iris localization scheme in a non-cooperative environment was realized. Then the normalization step can be realized by using the inner and outer boundary information of the iris, and the input of the iris verification deep learning architecture can be obtained. The primary findings of this study can be summarized as follows:
- We used a parallel circular transform to locate the inner boundary of the iris, and the fast Daugman integro-differential localization algorithm for locating the outer boundary, which results in effective and fast iris localization.
- A new lightweight deep learning architecture was developed for iris verification. The network architecture is based on ResNet, which is used to improve the robustness of the iris verification network architecture and the accuracy of the verification results.
- A residual pooling layer was introduced into the deep learning network architecture for iris recognition to extract more iris texture features.
The remainder of this paper is organized as follows: Section 2 introduces the related research on iris recognition. Section 3 presents details pertaining to the proposed iris localization and verification methods. Section 4 presents, compares, and analyzes the experimental results. Section 5 summarizes the research and concludes the study.
Point 10, Overall, Tables, and Figures should be explained comprehensively.
Response 10: we have improved the explain of figures and tables in the manuscript, Some examples as shown below:
Table 1 presents a comparison between the average running times of the serial and parallel algorithm for the iris localization stage. The implementation of the parallel algorithm of the proposed scheme on various databases reveals that the average speed-up on the CASIA.v4-distance, ND-LG4000, CASIA.v4-lamp, and CASIA-M1-S2 datasets is 26, 32, 36 and 21 times, respectively, indicating that the flexibility of the CUDA in performing parallel alignment accelerates the iris localization step. In actual experiments, circular Hough transform takes up most of the running time of the whole iris location, reaching more than 95%. Therefore, reducing the running time of circular Hough transform can effectively reduce the running time of the whole iris positioning. The experimental results confirm that the parallel algorithm of this scheme can effectively reduce the running time of the whole iris location.
The accuracy of iris localization is also an important indicator. Table 2 presents the accuracy of iris localization achieved using this method. The accuracy of iris location is the percentage of the number of correctly located iris to the total number of located iris. As shown in Table 2, the proposed method achieved good results on the four iris datasets collected in a non-cooperative environment, four different iris datasets all have a localization accuracy of more than 80%. To better comprehend the iris localization results, we selected certain challenging samples and presented the localization results obtained on the CASIA.v4-distance, ND-LG4000, CASIA.v4-lamp, and CASIA-M1-S2 datasets (Fig. 5). In Fig. 5, the inner and outer boundaries of the iris are marked by white circles, it is obvious that the iris location algorithm accurately extracts the inner and outer boundaries of the iris. We believe that this is due to the use of the iris location scheme proposed in this paper. The circular Hough transform and integro-differential operator are the mainstream iris inner and outer boundary location algorithms. The experimental results effectively confirm the advantages of these two algorithms.
The proposed iris verification method was evaluated on each database. To assess the effect of the residual pooling layer, several ablation experiments were conducted by ablating the residual pooling layer in the proposed network. The comparison results between iris recognition with and without the residual pooling layer are presented in Table 3 and Fig. 6, respectively. As shown in Fig. 6, on four datasets, with the same FAR, the FRR size of the proposed model is smaller than ResNet, indicating that the residual pooling layer has obvious advantages in improving iris verification accuracy. As shown in Table 3, on four datasets, the proposed network architecture outperforms 10.83%, 61.41%, 49.78% and 36.65% of FRR when the FAR was respectively, and 57.31%, 49.53%, 61.14% and 52.36% of ERR respectively, compared to the network architecture without the residual pooling layer.
As mentioned in the Materials and Methods section, a residual pooling layer is introduced into the proposed deep learning architecture. The residual pooling layer has the short connection module, the convolutional layer and the sigmoid function, which make the iris texture feature extraction more effective. Therefore, the corresponding ablation experimental results show that adding residual pooling layer can obtain higher iris recognition accuracy. On the other hand, compared with ResNet, the additional residual pooling layer increases the complexity of the deep learning model, which is the disadvantage of the deep learning architecture proposed in this paper.
The comparative experimental results obtained using the proposed method and the respective benchmark methods are depicted in Fig. 7. The EER and FRR, when the FAR was , obtained from the respective methods are summarized in Table 4. As shown in Fig. 6, on four datasets, with the same FAR, the FRR size of the proposed model is smaller than ResNet-18, MobileNet-v2 and Maxout-CNNs respectively, indicating that the proposed model has obvious advantages in improving iris verification accuracy.
Furthermore, the extent of performance improvement varies among the databases; the observed improvement is also supported by the EER and FRR@FAR= in the Table 4. The EER and FRR@FAR= of the proposed model is smaller than the other methods on the four iris datasets.
Maxout CNNs is a classic lightweight iris recognition deep learning architecture. Experiments on four datasets show that the iris recognition accuracy of the proposed depth learning architecture is higher than Maxout CNNs. We believe that this is because the proposed method uses the residual pooling layer and ResNet. More importantly, the network depth difference between Maxout CNNs and the proposed deep learning architecture is small, and we have selected the same super parameters for training, so in fact, the recognition accuracy beyond Maxout CNNs is not very obvious. ResNet-18 is an architecture widely used in various fields of image processing. The biggest difference between ResNet-18 and the proposed method is the lack of residual pooling layer, and the network depth is much higher than the proposed method. Too deep network is not conducive to iris recognition, because the number of iris data sets is relatively small compared with other large data sets, such as ImageNet. Therefore, the iris recognition accuracy of the method proposed in this paper is higher than ResNet-18. As an effective lightweight classical depth learning architecture, Mobile-v2 also performs well in iris recognition accuracy, which is only a little lower than the proposed method.
Point 11, The proper graph should be included for verifying the claimed performance such as the confusion matrix which is very necessary.
Response 11: Thanks for your thoughtful comments, we added the confusion matrix of experimental results in the manuscript, as shown below:
Fig. 8. illustrates the confusion matrices for iris verification obtained from the experimental results on the datasets of CASIA-Distance, ND-LG4000, CASIA-Lamp and CASIA-M1-S2, respectively. In the four iris datasets, all iris verified True Negative (TN) reached 99% (Green grid in the upper left corner). Since the number of genuine match scores is far less than the number of imposter match scores, the result of True Positive (TP) is lower than TN, but still higher than False Negative (FN) and False Positive (FP). Such high TN and TP indicate that the proposed model realizes high-precision iris verification.
Point 12, The results of real-time recognition of Chinese speakers should be included to accurately verify the testing performance of the proposed framework.
Response 12: This is a very good suggestion. In the next step, we will implement this framework on the embedded architecture and use the embedded system to realize real-time iris recognition of Chinese speakers.
Point 13, The conclusion of this research is very weak, the authors have not summarized the overall research work.
Response 13: we have rewritten the conclusion as follows:
In this study, we developed an efficient iris recognition system consist of iris localization and iris verification stage. To solve the problem of fast iris localization, we used a parallel circular transform to locate the inner boundary of the iris, after which the fast Daugman integro-differential localization algorithm was used to achieve effective and rapid iris localization. Furthermore, we propose a lightweight deep learning architecture with residual pooling layer to achieve high-precision iris recognition with less model parameters. The proposed iris localization method was implemented using 400 iris images obtained from four different representative iris databases, and the speed and effectiveness of this method were verified. The proposed iris verification method was evaluated using four representative iris image databases, and the performance of the deep learning architecture for high-precision iris recognition is verified. In general, the system can effectively realize iris recognition. First, the proposed positioning method is used to achieve fast and accurate iris localization stage, and then the proposed deep learning architecture is used to complete high-precision iris verification.
Although the proposed iris recognition scheme has many advantages above, it also has some defects. As there are two different schemes, the classical manual feature engineering method and the deep learning method, the problem is how to deploy the entire iris recognition process to the embedded device. Another problem is that the iris images we deal with are all illuminated under near infrared light. The iris localization scheme proposed in the paper can effectively extract the inner and outer boundaries of the iris, but for the irregular iris regions formed under visible light illumination, this scheme cannot achieve accurate iris localization. As for future work, we consider deploying the proposed system to the actual embedded architecture to achieve real-time iris recognition, and optimize the iris positioning algorithm to achieve iris accurate localization under visible lighting conditions.
Point 14, References are not enough and few, do not contradict and cover a specific era more literature should be added. References to newer or more recent work are required. But if authors pass this round, it is more meaningful to provide next time.
Response 14: We added a certain number of articles related to manuscripts as references, and the total number of references has been changed from 23 to 35. In the references, we cited the recent representative papers related to iris recognition based on depth learning.
Point 15, More recently published papers in the field of deep learning should be discussed in the Introduction/literature. The authors may be benefited by reviewing more papers such as: 10.3390/s21175892.
Response 15: We have discussed some articles in the field of deep learning in the introduction section, including the papers such as: 10.3390/s21175892.

Round 2
Reviewer 2 Report
The authors addressed my comments and suggestions successfully. Good Luck!